# Is Education Beneficial to Environmentally Friendly Behaviors? Evidence from CEOs

**DOI:** 10.3390/ijerph191811391

**Published:** 2022-09-10

**Authors:** Changrong Wang, Lufeng Gou, Xuemei Li

**Affiliations:** 1Department of Accounting, School of Business, Qingdao University, Qingdao 266071, China; 2Business School, Qingdao University of Technology, Qingdao 266520, China; 3School of Economics, Ocean University of China, Qingdao 266100, China; 4Institute of Marine Development, Ocean University of China, Qingdao 266100, China

**Keywords:** environmental impact, environmental protection investment, CEO education, CEO duality, manu industry, market competition

## Abstract

Corporate environmental investment decisions play a crucial role in the protection of the public environment. As the decision-maker and executor, the environmental consciousness and social responsibility of the chief executive officer (CEO) has a long-term impact on the company’s environmental protection strategy, and the CEO’s level of education is a significant factor influencing the CEO’s environmental protection decisions. In this paper, we investigate the extent to which CEO education influences environmental protection investment decisions. A CEO education index is constructed as a proxy for CEO education based on the CEO’s educational background, using a panel sample of Chinese listed firms from 2010 to 2019 and providing robust evidence supporting the notioin that firms with highly educated CEOs are likely to engage in environmental protection spending activities. However, the positive relationship between CEO education and corporate environmental protection investment is reduced when the CEO also holds the position of chairman. The heterogeneity analysis shows that the positive relationship between CEO education and corporate environmental investment behavior is stronger in non-manufacturing and highly monopolistic market competitive industries. Our study contributes to the sustainability literature by providing a new impetus for corporate environmental activities from the perspective of CEO education and sheds light on the impact of the internal and external factors of firms on the investment in environmental protection. It may also help decision makers to decide whether to hire highly educated CEOs and use a dual structure of CEOs in markets with different levels of competition.

## 1. Introduction

Governments and organizations are focusing on social practices and environmental rules for manage global warming because it is a serious concern for everyone and has become more prevalent over time [1]. The industrial sector is a significant contributor to environmental issues, particularly in developing economies [2]. Developed economies and emerging countries have raised concerns about environmental issues. The Chinese government started to focus on lowering pollution emissions and dealing with illegitimate emissions from businesses in 2015. The National People’s Congress and the Chinese People’s Political Consultative Conference proposed strategies and developed plans in 2015 to address the environmental pollution caused by air pollution, water pollution, and soil pollution [3,4]. China’s environmental protection policies were extensively rolled out between 2018 and 2019, and environmental protection measures were significantly bolstered. Market-oriented measures, such as environmental protection taxes and pollutant discharge permits, have been introduced [5,6,7].

Within this context, enterprises face tremendous pressure to protect the environment while achieving a growth strategy. Investment in environmental protection is a need that must be met if society and businesses are to accomplish sustainable development. The CEO is the principal decision-maker regarding corporate ecological protection investments. The literature on corporate governance has demonstrated that differences in firm performance and decisions can be explained by the human traits of CEOs [8,9]. When the management makes decisions, the CEO, who serves as the company’s chief decision-maker, will significantly influence the thoughts of the entire senior management team. 

Since CEOs have a crucial influence on their companies’ environmental policies, it is vital to examine the correlations between the educational background of CEOs and environmentally conscious business practices. The extant literature focuses almost exclusively on the role of the enterprise’s external environment or its internal governance mechanisms and how they influence the enterprise’s environmental protection investments and choices. For example, Xie et al. (2020) investigated environmental regulation subsidies [10]; Zhang et al. (2021) analyzed green policies [11]; Song et al. (2021) examined the impact of internal control [12]; Yang et al. (2021) focused on the supply chain quality [13]; and Wiengarten et al. (2013) examined environmental responsibility auditing behaviors [14]. Others examined how the personal attributes of CEOs are reflected in the level of corporate governance. For instance, Yan and Xu (2020) studied the impact of female directors [15], while Hu and Yang (2021) examined the political party groupings of executives [16].

To our knowledge, however, few studies have examined the impact of CEO education on corporate environmental investment by integrating the organization’s external environment and internal governance mechanisms. The level of education of corporate CEOs can influence business actions, and corporate executives with a higher level of education are more concerned with social responsibility issues [17,18] or corporate sustainable development problems [19]. CEOs believe that corporate social responsibility is a way for businesses to gain the favor of customers, investors, and the government, and thereby contribute to enhancing the company’s value [20,21,22]. Higher education causes CEOs to pay more attention to corporate social responsibility concerns and to be more inclined towards implementing environmental measures.

Although past research supports the link between CEO education and corporate social responsibility for a firm’s environmental strategy, no solid evidence has been found to support the notion that increased CEO education leads to company environmental protection spending. As a result, this study focuses on the following research questions: is higher CEO education beneficial for business environmental protection investment? If so, how do CEO duality (where the CEO also holds the position of board chairman), the market competitiveness level, and industry influence this relationship?

Following previous works in this field [23,24], based on Chinese listed firm data, we examine the relationship between CEO education and environmental investment in this study. We use the yearly increase in environmental protection spending by an enterprise to measure the environmental protection spending and the CEOs’ education degrees to measure the CEO educational level. We obtained 2710 research samples from 2010 to 2019 for the analysis. Firstly, we employed a fixed effect regression model (FE) to verify the main regression issues in our panel data. Secondly, we used a two-stage least squares (2SLS) regression analysis to overcome the endogeneity problems. Thirdly, we employed an alternative measure of the degree as a robustness test. Lastly, this study examined the effects of market competition and industry heterogeneity.

The main contributions of this paper to the study of the factors influencing corporate environmental strategies and environmental investment are as follows: (1) exploring the mechanism of the CEO’s educational level as an influencing factor of the environmental investment behavior of companies, this paper selects 2710 research samples from 2010 to 2019 and analyzes the relationship between the company CEOs’ educational level and the environmental investment behavior of companies using a panel model. (2) We investigate whether the power of CEO duality strengthens the relationship between the CEO’s educational level and corporate environmental protection investment. In existing firms in China, many corporate CEOs also serve as board chairs (CEO duality), which expands their power [25,26]. Based on agency theory and management theory, we further analyzed whether CEO duality affects the relationship between CEO education and corporate environmental protection investment. (3) A study was conducted to differentiate the relationship between the effects of market competition and industry heterogeneity on the level of CEO education and corporate environmental protection. Industry and market competition influence corporate decisions [27,28,29,30]. The degree of market competition affects firms’ investment and risk-taking [31,32], and industry characteristics can also influence firm decisions [33,34,35].

This study substantially complements and extends the existing literature regarding the role of CEO education and the moderating effect of CEO duality on the relationship between CEO education and enterprise green spending. We further discuss the essential functions of external factors, including market monopoly and industry type, to discover their influence on corporate environmental expenditure. Our research is innovative in three ways. Firstly, from the perspective of the CEO characteristics, we explore whether the CEO’s education can affect the company’s environmental protection investment behavior. Our research proves that CEO education is vital in increasing corporate environmental investment and optimizing corporate environmental behavior. Secondly, from the perspective of the corporate governance structure, we investigate the moderating effect of CEO duality on CEO education and corporate environmental behavior. Many scholars have proved that CEO duality impacts corporate policy and performance [36,37,38,39,40,41]. Our findings suggest that CEO duality weakens the positive relationship between CEO education and corporate environmental behavior. Thirdly, from the perspective of the macro-level factors, we discuss the external factors that drive environmental protection investment. The integration of micro- and macro-level factors in this analysis was expected to provide a more nuanced and comprehensive understanding of the driving forces behind corporate environmental investment behaviors.

The remainder of this paper proceeds as follows. Section 2 examines the theoretical background and develops our hypothesis. Section 3 describes the research design. Section 4 reports and discusses our empirical results. Section 5 addresses the additional analysis. Section 6 introduces our conclusions.

## 2. Theoretical Background and Hypothesis Development

### 2.1. CEO Education and Environmental Protection Investment Decisions

In this paper, the primary research on corporate social responsibility activities refers to the meeting of stakeholders’ expectations and the helping of enterprises to achieve a sustainable performance through the optimization of environmental protection investment activities. We mainly explore whether the green investment behavior of enterprises is affected by the educational level of their CEOs. Consequently, taking environmental protection investment as one of the company’s socially responsible activities, we hypothesize:

**Hypothesis** **1.** 
*CEO education is positively related to corporate environmental protection investment decisions.*


A person’s education level can strongly influence his or her cognitive model [42]. Education can provide societal benefits [43]. Based on upper echelon theory, CEOs’ educational level affects their decisions. CEOs’ professional knowledge affects their understanding and ability [44]. CEOs with higher educational levels are more sensitive to new ideas and are willing to take risks [45,46]. More highly educated people care more about society, nature, and the environment. They have a higher environmental awareness and prefer implementing sustainable development strategies through the green management of companies [47]. Scholars suggest that the CEO is critical in launching a company’s environmental initiatives. The CEO’s decision is crucial for implementing environmental practices that improve the company’s environmental performance [48]. The formal educational level and environmental expertise of the CEO are closely related to the company’s higher compliance with green initiatives and environmental performance rating [49]. It is known that the higher the educational level of the CEO is, the more he/she will promote the environmental innovation behavior of the enterprise [50]. The CEO’s education has a significant role in promoting the energy efficiency of the enterprise. It has been proved that CEO education is associated with greater environmental awareness. Highly educated CEOs exhibit more essential concerns about climate change and drive more environmentally efficient cars [47]. Female CEOs and the CEO educational level are related to the likelihood of a positive corporate environmental performance [51].

In addition, the more educated the CEO is, the more willing he or she is to consider corporate social responsibility (CSR) issues when setting the corporate strategy. This is not only because education improves the green awareness of CEOs and enables CEOs to combine the performance strategy with a sustainable development strategy when formulating corporate strategies, but also because many enterprises view the performance of social responsibility as a means of winning customers’ favor and establishing a positive image and reputation [52]. Through higher-level educational courses and training, CEOs learn the importance of CSR for a business, and they come to understand how significant the pursuit of CSR activities by a company is for winning the potential customer’s favor and attaining economic interest. They view CSR activities and spending as a marketing opportunity rather than an investment project [53,54]. Therefore, more highly educated CEOs are more likely to take further environmentally friendly actions so as to increase their profits and attract more potential customers. Based on stakeholder theory, corporate social responsibility performance is based on different stakeholders [55]. The environmental protection behavior through CSR activities aims to reduce the damage to the natural environment by reducing the pollution discharged by enterprises and controlling all kinds of pollution, such as wastewater pollution, solid pollution, and noise pollution [56].

### 2.2. Moderating Effects of CEO Duality

It is common for a person to act as both the chairman and CEO of an enterprise in China. Because the professional manager system in China’s market is not perfect, this phenomenon of CEO duality is widespread in China’s private enterprises. According to the research conclusions of previous scholars, it is unknown whether CEO duality is beneficial for companies in the long run [28,57,58,59].

This study adopts the perspective of agency theory to consider the mediating effects of CEO duality on CEO education and enterprise environmental protection investment. When the chairman and CEO positions are held by the same person, the CEO is more likely to use his/her centralized power to achieve his/her personal goals at the expense of the organization’s interests. Hence, the duality of the CEO and chairman may negate the positive relationship between executive education and the company’s environmental protection investment. Therefore, we propose the following Hypothesis 2:

**Hypothesis** **2.** 
*Ceteris paribus, CEO duality moderates the positive effect of CEO education on corporate environmental protection investment decisions.*


The academic research on CEO duality mainly focuses on the relationship between CEO duality and corporate performance, but the conclusions thus far are controversial. CEO duality has positive and negative effects on corporate governance [46]. Based on agency theory, when the power given to professional managers is too high and weakens the power of the board of directors in providing supervision, it provides CEOs with the opportunity to sacrifice stakeholders’ benefits in order to obtain their own benefits [57,58,59,60]. Therefore, based on agency theory, CEO duality it is not advisable, as it weakens corporate governance [59]. However, stewardship theory asserts that CEO duality can ensure the concentration of power and the confidence of managers, which may positively impact the speed of enterprise decision-making, lower costs, and improve efficiency and enterprise performance [58,61,62,63].

Additionally, Jianyun [64] has shown that, when the CEO has more power than the other executives, and when there is a controlling external director on the board of directors, the impact of CEO duality on enterprise performance is negative. Moreover, Kamarudin, et al. [65] found that earning quality is positively related to the independence of the audit committee, but this relationship is weakened by CEO duality. Uyar, et al. [66] suggested that, depending on the CSR dimension, CEO duality moderates the association between CSR performance and tourism sector development. According to Alves’s study [67], CEO duality reduces the earning quality. Furthermore, these findings suggest that, when the board of directors includes a higher proportion of independent directors, the earning quality reduction associated with CEO duality is mitigated. According to Wijethilake and Ekanayake [68], CEO duality hinders the firm performance when the CEO has additional informal power. In contrast, CEO duality favors a company’s success when the board is heavily involved. The abovementioned research shows that CEO duality impacts corporate performance and earning quality in relation to mediating or intermediary variables. In this paper, we further investigate the mediating impact of CEO duality on the relationship between CEO education and enterprise environmental protection behavior based on the existing research of other scholars.

## 3. Models, Variables and Data

### 3.1. Sample Selection and Data Sources

This paper takes Chinese A-share-listed companies from 2010 to 2019 as the research sample. We exclude financial companies, ST-listed companies, and samples with missing data on environmental protection investment. The CEO education, CEO duality, environmental protection investment, and data on other control variables used in this paper were obtained from the CSMAR database, Wind database, and firms’ financial statements. All variables were winsorized at the 1st and 99th percentiles to avoid the effects of extreme values. Eventually, 2710 research samples were obtained.

### 3.2. Measurement of the Key Variables

#### 3.2.1. Dependent Variable: Corporate Environmental Protection Investment

Some scholars prefer to use environmental protection data from corporate social responsibility reports or environmental responsibility reports as proxy variables in the analysis of corporate environmental protection investment [69], but this method is highly subjective. Referring to the existing methods used by scholars, to reduce the influence of subjectivity, we chose to extract data from the notes of firms’ financial statements [23,70]. We extracted keywords related to the enterprises’ environmental protection and governance policies regarding the construction work in progress, general and administrative expenses, and other payables in the notes of the financial statements of the listed companies. The keywords for the extraction and screening included cleaning, greening, environmental protection, pollution discharge, energy-saving, carbon dioxide, emission reduction, etc. Since an independent third party had audited the financial statements of the listed companies, measuring corporate environmental protection investment by extracting environmental-protection-related keywords from the notes of the financial statements enables the environmental protection investment data to more objective and authentic. The yearly increase in environmental protection spending was taken as the proxy variable of environmental protection investment. In order to improve the stability of the data, we used the natural logarithm of the yearly increase in environmental protection investment by enterprises as the dependent variable [13,22,71,72].

#### 3.2.2. Independent Variable: CEO Education

Due to the differences in the level of executives’ education, the investigation of on-the-job education and academic education is usually controversial. Through the CSMAR database and manual collection of information disclosed in enterprise financial statements, we followed Ting et al. [73] and Rakhmayil and Yuce [74] by using a scale range of 1 to 5 to distinguish the educational level of the CEOs and divide the educational level of the CEOs into the following levels: 1 = technical secondary school or below, 2 = junior college, 3 = bachelor’s degree, 4 = master’s degree, MBA/EMBA, and 5 = doctoral degree or above. Many researchers employ scale ranges and numbers to represent CEO education, with a higher value indicating a higher level of education received by the CEO [23,75,76]. In the robustness test, we condensed the level of CEO education. Follow Huang [19] and Chithambo, et al. [77], we defined whether the CEO had obtained a bachelor’s degree or above as the proxy variable for whether he/she had received higher education and generated a dummy variable. If the CEO had obtained a bachelor’s degree or above, we created a dummy variable 1. If the CEO did not have a bachelor’s degree or below, we set the dummy variable as 0 [19,77].

#### 3.2.3. Moderating Variable

CEO duality was chosen as the moderating variable and a dummy variable. It was equal to 1 when a firm’s CEO also held the board chairman position; otherwise, it was equal to 0 [78].

#### 3.2.4. Control Variables

We included the following CEO characteristics and firm control variables in our regression mode to control other variables affecting the companies’ environmental protection investment decisions. The control variables for the CEO characteristics included CEO gender (GENDER) and CEO age (AGE), as existing research indicates that CEO behavior varies between age groups [79,80]. Moreover, gender-diverse management teams are more likely to implement environmental strategies [81]. The firm variables included variables that have the potential to affect corporate environmental protection investment decisions, including manu-industry status (MANU), stock ownership (OWNERSHIP), dispersion (DISPERSION), leverage (LEV), return on assets (ROA), firm performance (Tobin Q), cash-to-profit ratio (CASH), and industry-sector effects (INDSEC).

Enterprises that manufacture products are more likely to emit pollutants and pollute the environment than non-manufacturing industries [82]. Therefore, whether firms belong to the manufacturing industry is an essential factor affecting enterprises’ environmental protection investments. Therefore, we chose the industry as one of the control variables. In this paper, we generated dummy variables. If the firm belonged to a manufacturing industry company, we generated it as 1; otherwise, it was 0. We also selected state ownership as a control variable [83,84,85]. Because of the particularity of state-owned enterprises in China, the absence of shareholders reduces the role of the board of directors and is likely to increase the power of the CEO himself/herself, which may affect the company’s environmental investment decisions. We set the sample enterprise as 1 if it was a state-owned enterprise and 0 if it was a private enterprise.

Similarly, corporate performance, profit margins, and cash holdings may also affect corporate environmental decisions. We selected corporate ROA and Tobin Q to represent the corporate financial performance [86]. Cash-to-profit was chosen to represent the cash-to-profit ratio of the enterprise. Dispersion was selected as the proxy variable for the separation of an enterprise’s rights because it affects the CEOs’ decisions and relates to the difference between the cash flow rights and controllers’ voting rights [87,88]. Leverage was chosen as one of the control variables because creditors play the role of governance, which contributes to the company’s green decisions and performance [89]. It is measured as the total debt scaled by the total assets. Table 1 displays the descriptions and measurements of these variables.

### 3.3. Models Specification

We conducted Hausman tests, and the Hausman test results show that all *p* values were less than 0.001. Therefore, we chose the fixed effects approach to construct the regression model [90]. We used the following panel regression model 1 to test the relationship between the CEO education and corporate environmental protection investment:(1)$$\begin{array}{c} \;Env_{i,t}=\beta_{0}+\beta_{1}Degree_{i,t}+\beta_{2}Gender_{i,t}+\beta_{3}Age_{i,t}+\beta_{4}Manu_{i,t}+\beta_{5}SOE_{i,t}+\beta_{6}Dispersion_{i,t}+\\ \beta_{7}Lev_{i,t}+\beta_{8}Roa_{i,t}+\beta_{9}TobinQ_{i,t}+\beta_{10}Cash_{i,t}+\beta_{11}Industry+\varepsilon_{i,t} \end{array}$$

In addition to the baseline model, we also examined the moderating effects of CEO duality on the relationship between the CEO’s educational degree and corporate environmental protection investment. We created model 2 and introduced the interaction term of CEO duality and CEO degree (Dual × Degree), and *β*_3_ of the following model 2 was used to measure the moderation effects:(2)$$\begin{array}{c} \begin{array}{l} \;Env_{i,t}=\beta_{0}+\beta_{1}Degree_{i,t}+\beta_{2}Dual_{i,t}+\beta_{3}Dual\times Degree_{i,t}+\beta_{4}Gender_{i,t}+\beta_{5}Age_{i,t}+\beta_{6}Manu_{i,t}+\beta_{7}SOE_{i,t}\\ +\beta_{8}Dispersion_{i,t}+\beta_{9}Lev_{i,t}+\beta_{10}Roa_{i,t}+\beta_{11}TobinQ_{i,t}+\beta_{12}Cash_{i,t}+\beta_{13}Industry+\varepsilon_{i,t} \end{array} \end{array}$$

## 4. Empirical Results

### 4.1. Descriptive Statistics

From the descriptive statistics in Table 2, we can observe a great difference in the environmental protection investment activities of different enterprises. The natural logarithm of the highest increased environmental protection investment amounts to 21.54, and the lowest value is 11.61, which also shows the attitude of enterprises towards environmental protection investment. There are significant differences which also lay the foundation for our research. At the same time, we can see that the educational background of CEOs varies greatly. Some CEOs have a degree below a college degree, and some have a doctoral degree, which indicates that our research is highly meaningful.

From Panel B, above, it can be seen that the increase in the amount of environmental protection investment by Chinese companies has been increasing each year. The total amount of the environmental protection investments of the sample companies increased from 16,530 million Yuan in 2010 to 37,830 million Yuan in 2019, and it more than doubled in ten years.

### 4.2. Pearson Correlation Analysis

The Pearson pairwise correlation matrix is presented in Table 3. We found a positive correlation between CEO’s educational level and corporate environmental protection investment at a significant level of 1%, which also preliminarily verified our Hypothesis 1. The more educated the CEO is, the more likely the company is to invest in environmental protection. Moreover, we found that the Lerner index and corporate environmental protection investment have a significant positive relationship of 0.088, which is significant at 1%, preliminarily indicating that businesses are more likely to engage in environmental protection investment practices when the market environment is more monopolistic. In addition, our corporate environmental protection investment measurements and all the control variables are highly positively (negatively) related, indicating that it was appropriate to control these variables in our regression model. These variables include manu (MANU), stock ownership (OWNERSHIP), dispersion (DISPERSON), leverage (LEV), return on assets (ROA), firm performance (Tobin Q) and cash-to-profit ratio (CASH).

Finally, we computed the variance inflation factors (VIFs) when estimating our baseline regression model to test for signs of multicollinearity. Overall, we found that none of the VIFs exceeded 5 and concluded that this model is not affected by multicollinearity problems [91].

### 4.3. Baseline Regression Results for Education Background, Duality, and Corporate Environmental Protection Investment

From Panel A in Table 4, it can be seen that, without adding the control variables, there is a positive correlation between CEO education and corporate environmental protection investment in column (1), with a coefficient of 0.1428, which is significantly positive at the 10% level, and this proves our hypothesis 1. Adding the control variables, the positive relationship between CEO education and corporate environmental protection investment in column (2) is strengthened, with a coefficient of 0.1998, which is significantly positive at the 1% level. This proves our Hypothesis 1, again, and indicates that the improvement of CEOs’ educational levels can promote environmental protection investment by enterprises. However, from the results of column (3) in panel A of Table 3, it can be seen that CEO duality does not affect the company’s environmental protection investment behavior. When we multiply the CEO education and CEO duality in column (4) (Dual × Degree), we discover that the coefficient of the interaction term between CEO duality and CEO education (Dual × Degree) is −0.3193, which is significantly negatively correlated at the 10% level. This result reveals that CEO duality diminishes the positive relationship between CEO education and corporate environmental protection, proving Hypothesis 2 of this paper, that is, that CEO duality weakens the positive impact of CEO’s education on corporate environmental protection investment. When the CEO is also the chairman, his or her power may grow. As a result, the CEO may be more likely to use that power to pursue a personal goal, such as increasing short-term corporate earnings for pay compensation. Investment in environmental protection increases costs and decreases profitability. Therefore, CEO duality diminishes the favorable effect of CEO education on corporate environmental protection spending.

Panel B of Table 4 shows the heavily polluting enterprises that were selected for further analysis. The selection of high-polluting industries was mainly based on the “Environmental Inspection List of Listed Companies” and “Guidelines for Environmental Information Disclosure of Listed Companies” formulated by the Ministry of Environmental Protection of China in 2008 and 2010. The list mainly includes 16 highly polluting industries, such as coal, mining, textile, leather, paper, petrochemical, pharmaceutical, chemical, metallurgy, thermal power, etc. Through screening, we found that 1779 sample enterprises among the entire sample belong to highly polluting enterprises. Panel B shows the regression results of highly polluting companies. According to Panel B, there is still a positive relationship between CEO education and corporate environmental protection investment. In columns (2), (3), and (4), the regression coefficients for the influence of CEO education on company environmental protection investment are 0.2713, 0.2720, and 0.3408, respectively, and they are all significantly correlated at the 1% level. This result is consistent with our Hypothesis 1. We can see that the influence of CEO education on corporate environmental protection investment is greater in Panel B (the highly polluting enterprise sample) than in Panel A (the full sample), indicating that CEO education may play a more influential role in highly polluting enterprises in regared to the promotion of enterprise environmental investment.

In addition, in column (4) of Panel B, CEO duality in highly polluting companies negatively affects the relationship between CEO education and environmental protection investment. Hypothesis 2 is proved again. Compared with the regression result of Panel A of the whole sample, in Panel B, the coefficient of the interaction term (Dual × Degree) of highly polluting enterprises is −0.4309 in column (4), which is significantly negatively correlated at the level of 5%. The result shows that, in heavily polluting enterprises, CEO duality has a more substantial negative moderating effect on the relationship between corporate CEO education and environmental protection investment.

### 4.4. Robustness Checks for Endogeneity Using the Two-Stage Least Squares (2SLS) Method

To prevent the endogeneity of the results, we tested the results through the 2SLSmethod. As shown in Table 5 and Table 6, we verified Hypothesis 1 and Hypothesis 2 of this paper through further tests.

We performed a 2SLS regression to address the potential endogeneity between CEO education and corporate environmental protection spending. According to previous literature [92], we defined whether the CEO was born during China’s Cultural Revolution as an instrumental variable for the CEO education [93,94]. China’s Cultural Revolution lasted from 1966 to 1976. Therefore, if CEOs were born in 1948–1958, so that they were 18 years old during China’s Cultural Revolution (1966–1976), they had fewer opportunities to go to college [94]. We created a dummy variable degree instrument to represent whether the CEO was born from 1948–1958. We used this instrumental variable to represent whether the CEO had a college degree [94]. This instrumental variable was highly correlated with our explanatory variable, CEO education, but had no direct impact on corporate environmental protection investment.

We used CEO education as the dependent variable to build the model in the first stage of the analysis. The results show that the degree instrument (whether the CEO attained the age of 18 during China’s Cultural Revolution) was negatively correlated with CEO education, with a coefficient of −0.3398 at the 1% significance level. This suggests that if a CEO was 18 during the Cultural Revolution, his/her education level would be affected. China’s Cultural Revolution had a significant adverse effect on the CEOs’ education.

In the second stage, the amount of corporate environmental protection investment was used as the dependent variable. From the results of the second stage in Table 3, it is shown that the CEO’s education is positively related to corporate environmental protection investment and is significant at 1%, indicating that the higher the educational level of the CEO is, the more likely it is that the company will engage in environmental protection investment. This, again, proves Hypothesis 1. We also found a significant negative correlation between (Dual × Degree) and the dependent variable. The regression coefficient was −1.0757, which was significantly correlated at the level of 1%. This shows that CEO duality can significantly weaken the positive relationship between CEO education and corporate environmental protection investment. Again, this verfieis Hypothesis 2.

The F-value of the first stage is 234.48, and Shea’s partial R-squared for the first-stage model is 0.0170 and below 0.05. The F-value and Shea’s partial R-squared show that the instruments are correlated with the potentially endogenous variable [95]; thus, there is no problem affecting the weak instrumental variables.

### 4.5. Robustness Check for the Alternative Measure of the CEO Degree

We used degree 2 as an alternative measure of the degree. If the CEO had a bachelor’s degree or above, we set the degree 2 as 1; otherwise, it was 0 [96]. The regression results are consistent with the results of the main regression analysis. Panel A in Table 7 shows the full sample regression results. When variable degree 1 is replaced with variable degree 2, we can see that degree 2 and the enterprise environmental protection investment have a significant and positive relationship. The interaction term of duality and degree 2 is further correlated, and the coefficient of the interaction term demonstrates a substantial negative correlation, which, again, proves Hypothesis 2. Panel B shows the regression results of the sample of polluting enterprises. The findings are consistent with the previous results, demonstrating the robustness of the major analytical conclusions.

## 5. Additional Analysis

### 5.1. Heterogeneity Effects of the Manufacturing Industry and Non-Manufacturing Industry

When enterprises belong to different industries, their environmental protection investment decisions are different [97]. To facilitate their comparison, we further divided the enterprises into the manufacturing and non-manufacturing industries. Table 8 shows a comparison between the manufacturing enterprises and non-manufacturing enterprises. Column (1) and column (2) indicate the degree, selected as as the independent variable, and the dependent variable is the natural logarithm of the amount of corporate environmental protection investment. Degree 2 (whether the CEO has a bachelor’s degree) was used as a dependent variable, as seen in column (3) and column (4). Column (1) and column (3) show the regression results of the manufacturing enterprises, and column (2) and column (4) show the regression results of the non-manufacturing enterprises.

In non-manufacturing companies, CEO education has a more significant impact on environmental protection investment. The regression coefficients of CEO education in column (2) and column (4) are 0.3728 and 1.6352, respectively, which are greater than the coefficients in column (1) and column (3). The regression coefficient of CEO education is significantly correlated at 5%. 

The reasons for the manufacturing industry’s investsment in environmental protection may be relate to the mandatory requirements of the national environmental protection policy or the government’s constraints on environmental protection emissions. Therefore, subjective executive characteristics, such as CEO education, have less effect on corporate environmental protection investment behaviors. 

### 5.2. Heterogeneity Effects by Market Competition

Table 9 shows the impact of the market competition intensity on the relationship between CEO education and corporate environmental protection investment. In this paper, we use the Lerner index to measure the market’s competition intensity. A high Lerner index value indicates that the industry has low competition (high monopoly). Panel A shows the full sample results. Firstly, we calculated the mean of the Lerner indexes of the sample firms. After comparing the result with the mean of the Lerner index, we divides the sample firms into a low-competition group (Panel B) and a high-competition group (Panel C).

We found that the coefficients of the Lerner index of column (1) and column (2) were 2.6145 and 2.583 and significant at 5%. This shows that the lower the market competition of the industry is (i.e., the higher the degree of monopoly), the more likely it is that the companies in the industry will prefer to engage in environmental protection investments. This may be because enterprises are more likely to invest in environmental protection when subject to mandatory policies. Therefore, the lower the market competition is, the more attention they receive from the government, and the more likely they are to be encouraged or compelled to increase their environmental protection investment.

In addition, we also found that the regression coefficients of CEO education in column (3) and column (4) were 0.2898 and 0.2082 in the low-competition group and high-competition group, respectively, and the significance was at the level of 5%, which shows that, in a higher-monopoly industry, the positive effect of CEO education on environmental protection investment is stronger. It also shows that the monopoly of the market, in terms of competition, promotes and positively moderates the relationship between CEO education and the firm’s environmental protection investment.

## 6. Conclusions

Using the panel data of the listed companies in China from 2009 to 2019, this paper empirically analyzes the impact of CEO education on corporate environmental protection investment behavior and the mediating effects of CEO duality on the relationship between CEO education and firms’ environmental protection investment behaviors. The results show that: (1) CEO education plays a positive role in promoting the firms’ environmental protection investment behaviors, after controlling certain control variables. (2) CEO duality can be used as a moderating variable of the relationship between CEO education and firms’ environmental protection investment behaviors. The model results show that CEO duality weakens the positive relationship between CEO education and firms’ environmental protection investment behavior. (3) When the research samples are divided into non-manufacturing and manufacturing enterprises, the heterogeneity results show that CEO education has a greater impact on firms’ environmental protection investment behavior in non-manufacturing enterprises. This means that, although the characteristics of corporate executives influence corporate environmental protection investment behaviors, external policies and the environment also significantly impact corporate environmental protection investment. At the same time, the manufacturing industry is more affected by external factors, such as policies and government pressure to invest in environmental protection upgrades. Therefore, non-manufacturing enterprises are driven more by the characteristics of the enterprise’s internal executives to invest in environmental protection. (4) When dividing the research samples into a high-market-competition group and low-market-competition group, according to the market competition environment, the results show that the degree of market competition can affect the environmental protection investment by enterprises, and the increase in the market monopoly can promote the environmental protection investment by enterprises. Our further analysis demonstrates that, when market competitiveness is low, the favorable effect of CEO education on company environmental protection investment behavior can be enhanced.

Our results support the upper echelons theory [44]. Executives’ traits influence companies’ decisions, including social responsibility decisions [18,77]. As the primary decision-maker of the enterprise, the CEO directly affects the social responsibility decision-making of the enterprise [98,99]. Corporate executives of a higher educational level are more inclined to fulfill their social responsibility duties. Our research results demonstrate that improving the CEO’s educational level can encourage environmental protection investment and upgrading. In addition, based on stewardship theory, the concentration of power brought by CEO duality negatively moderates the role of CEO education in promoting the firms’ environmental protection investment behaviors. Therefore, whether the CEO has a high level of education and whether CEO duality is selected as the management model should be regarded key elements affecting enterprises’ decisions regarding the CEO candidates from the standpoint of the encouragement of the upgrading of the enterprises’ environmental protection activities.

In the future, on the one hand, the measurement of environmental protection investment requires further study. Currently, because of the difficulty of obtaining data directly from the database or the main sections of the financial statements, most scholars manually collect data from the notes of the financial statements in order to measure the amount of environmental protection spending.

On the other hand, due to limits relating to the information disclosure policies and databases of Chinese enterprises, the education of enterprise executives is only measured by the level of education that they have received. There is no further classification, such as which school the CEO graduated from, or whether they attended an ordinary university or a top-level university. Furthermore, questions related to the degree types of CEOs, such as whether CEOs with a high-level degree in environmental or natural sciences are more likely to engage in environmental protection than CEOs with a degree in management economics or engineering, should be the subject of further research.

Based on upper echelons theory and theoretical and empirical analyses, this paper discusses the impact of CEO education on enterprise environmental protection investment activities. We provide several suggestions for studying the economic consequences of CEO education and expand the research on the motivators of environmental protection investment activities. Research on CEO education can be further refined based on the type of university and the academic specialization of the CEO. In addition, in the future, we will further study the impact of environmental protection investment on corporate performance according to different CEO education level.

## Figures and Tables

**Table 1 ijerph-19-11391-t001:** The descriptions and measurements of the variables.

Variables	Full Name	Definitions
Env	Environmental protection investment	Natural logarithm of the increase in corporate environmental protection investment in the current year.
Degree	CEO degree	CEO’s educational background: 1 = technical secondary school or below, 2 = college, 3 = undergraduate, 4 = master’s degree and MBA/EMBA, 5 = doctoral degree and above.
Gender	CEO gender	Dummy variable, coded as 1 if the CEO is male, and 0 otherwise.
Age	CEO age	CEO’s age.
Dual	Duality	Dummy variable, coded as 1 if the CEO is also chairman, and 0 otherwise.
Manu	Whether it is a manu-industry company	Dummy variable, coded as 1 if the company belongs to the manufacturing industry and 0 otherwise (based on the industry classification guidelines for the listed companies issued by the China Securities Regulatory Commission in 2012).
SOE	Ownership	Dummy variable, coded as 1 if the company is a state-owned company, and 0 otherwise.
Dispersion	Dispersion	The difference between the cash flow rights and ownership of the actual controller.
Lev	Leverage	Total debt scaled by total assets
Roa	Return on assets	Net profit/total assets.
Tobin Q	Tobin Q	Tobin’s Q = market value/total assets.
Cash	Cash-to-profit ratio	Cash/total profit.
Lerner index	Lerner index	(Company operating income/total operating income in the industry) × cumulative Lerner index of individual stocks.
Industry	Industry	Dummy variable, coded as 1 if the firm is represented in a particular CSRC category, and 0 otherwise.

**Table 2 ijerph-19-11391-t002:** Descriptive statistics.

Panel A: Descriptive Statistics for All Variables					
Variable	Obs	Mean	Std. Dev.	Min	Max
Env	2710	16.623	2.092	11.612	21.541
Degree	2699	3.472	0.839	1	5
Gender	2710	0.945	0.228	0	1
Age	2710	49.941	5.868	35	66
Dual	2710	0.137	0.344	0	1
Manu	2710	0.773	0.419	0	1
SOE	2710	0.631	0.483	0	1
Dispersion	2708	0.059	0.085	0	0.309
Lev	2710	0.497	0.198	0.078	0.95
Roa	2710	0.03	0.101	−2.16	2.163
Tobin Q	2642	1.777	1.039	0.849	6.946
Cash	2710	1.593	4.108	−13.076	21.952
Lerner index	2737	0.112	0.077	0.009	0.412
**Panel B: Descriptive Statistics for Environmental Protection Investment**					
**Year**	**Sum**	**Mean**	**SD**	**Min**	**Max**
2010	16,530,000,000.00	60,324,865.00	207,124,986.98	110,389.01	2,266,000,000.00
2011	21,470,000,000.00	78,635,977.00	242,702,511.07	110,389.01	2,094,000,000.00
2012	26,220,000,000.00	96,032,666.00	295,967,631.94	110,389.01	2,266,000,000.00
2013	30,180,000,000.00	110,200,000.00	300,005,890.67	110,389.01	2,266,000,000.00
2014	35,890,000,000.00	131,000,000.00	351,695,446.14	141,515.98	2,266,000,000.00
2015	33,330,000,000.00	121,700,000.00	320,528,579.49	110,389.01	2,266,000,000.00
2016	32,480,000,000.00	119,000,000.00	292,022,822.57	120,000.00	2,266,000,000.00
2017	34,130,000,000.00	124,600,000.00	328,658,839.57	110,389.01	2,266,000,000.00
2018	38,490,000,000.00	140,500,000.00	345,840,620.13	110,389.01	2,266,000,000.00
2019	37,830,000,000.00	138,100,000.00	326,224,633.14	110,389.01	2,266,000,000.00

**Table 3 ijerph-19-11391-t003:** Pearson correlation matrix.

	Env	Degree	Gender	Dual	Lerner Index	Age	Manu	SOE	Dispersion	Lev	ROA	Tobin Q	Cash
Env	1												
Degree	0.139 ***	1											
Gender	0.062 ***	0.061 ***	1										
Dual	−0.03	−0.093 ***	−0.111 ***	1									
Lerner index	0.088 ***	0.028	−0.025	0.001	1								
Age	0.108 ***	−0.142 ***	−0.017	0.179 ***	0.109 ***	1							
Manu	−0.043 **	−0.103 ***	−0.015	0.101 ***	−0.182 ***	−0.152 ***	1						
SOE	0.134 ***	0.112 ***	0.094 ***	−0.208 ***	0.037 *	0.103 ***	−0.204 ***	1					
Dispersion	0.077 ***	0.001	0.092 ***	−0.106 ***	0.058 ***	0.022	0.044 **	−0.083 ***	1				
Lev	0.268 ***	0.102 ***	0.005	−0.048 **	−0.135 ***	0.078 ***	−0.151 ***	0.238 ***	0.059 ***	1			
ROA	−0.011	−0.038 **	−0.027	0.025	0.125 ***	−0.001	0.028	−0.077 ***	0.036 *	−0.226 ***	1		
Tobin Q	−0.246 ***	−0.100 ***	0.03	0.072 ***	0.052 ***	−0.025	0.037*	−0.122 ***	−0.019	−0.244 ***	0.061 ***	1	
Cash	0.085 ***	−0.002	−0.023	−0.035 *	−0.018	0.02	−0.003	0.046 **	0.033 *	0.097 ***	−0.021	−0.090 ***	1

Note: this table displays the Pearson correlation coefficients of CEO education, company environmental protection investment, and the control variables. Significance levels of 10%, 5%, and 1% are denoted by *, **, and ***, respectively. The definitions of the variables are provided in Table 1.

**Table 4 ijerph-19-11391-t004:** Baseline regression results: CEO education, duality, and corporate environmental protection investment.

	Panel A: Regression Results for the Full Sample			
		Fixed Effect Regression		
	Coefficient (*t*-Statistic)	Coefficient (*t*-Statistic)	Coefficient (*t*-Statistic)	Coefficient (*t*-Statistic)
	(1)	(2)	(3)	(4)
	Env	Env	Env	Env
Degree	0.1428 *	0.1998 ***	0.2011 ***	0.2468 ***
	(1.8183)	(2.6574)	(2.6795)	(3.0068)
Dual			−0.1346	0.9518
			(−0.8788)	(1.5103)
Dual × Degree				−0.3193 *
				(−1.8571)
Gender		0.0172	0.0008	−0.0216
		(0.0560)	(0.0026)	(−0.0717)
Age		0.0274 ***	0.0292 ***	0.0290 ***
		(2.8990)	(2.9833)	(2.9636)
Manu		−1.7716 ***	−1.7875 ***	−1.7868 ***
		(−7.0798)	(−7.1621)	(−7.0323)
SOE		−0.3635	−0.3708	−0.3730
		(−1.5116)	(−1.5522)	(−1.5252)
Dispersion		0.6908	0.6826	0.6685
		(0.6585)	(0.6494)	(0.6346)
Lev		0.4218	0.4116	0.3988
		(0.9975)	(0.9784)	(0.9434)
ROA		−0.8834	−0.8790	−0.9049
		(−1.0886)	(−1.0859)	(−1.1337)
Tobin Q		−0.1429 **	−0.1433 **	−0.1423 **
		(−2.2982)	(−2.3130)	(−2.3206)
Cash		−0.0103	−0.0105	−0.0102
		(−1.6118)	(−1.6473)	(−1.6020)
_Cons	16.1203 ***	16.2951 ***	16.2570 ***	16.1323 ***
	(59.0876)	(23.6746)	(23.7840)	(23.1242)
Industry	Yes	Yes	Yes	Yes
*N*	2726	2649	2649	2649
Adj. *R*^2^	0.0032	0.0185	0.0186	0.0212
**Panel B: Regression Results for the Heavy Pollution Companies**				
	**Fixed Effect Regression**			
	**Coefficient** **(*t*-Statistic)**	**Coefficient** **(*t*-Statistic)**	**Coefficient** **(*t*-Statistic)**	**Coefficient** **(*t*-Statistic)**
	**(1)**	**(2)**	**(3)**	**(4)**
	**Env**	**Env**	**Env**	**Env**
Degree		0.2713 ***	0.2720 ***	0.3408 ***
		(3.0821)	(3.0961)	(3.5731)
Dual			−0.0638	1.3884 **
			(−0.3693)	(2.0490)
Dual × Degree				−0.4309 **
				(−2.3370)
Gender	0.1044	0.0530	0.0469	0.0133
	(0.2411)	(0.1237)	(0.1096)	(0.0315)
Age	0.0195*	0.0296 ***	0.0305 ***	0.0293 ***
	(1.7869)	(2.8558)	(2.8173)	(2.7118)
Manu	0.0000	0.0000	0.0000	0.0000
	(.)	(.)	(.)	(.)
SOE	−0.9226 ***	−0.8828 ***	−0.8876 ***	−0.8976 ***
	(−3.1625)	(−2.8437)	(−2.8869)	(−2.8399)
Dispersion	−1.0223	−0.9310	−0.9370	−1.0095
	(−0.9353)	(−0.8195)	(−0.8235)	(−0.8897)
Lev	0.2937	0.1526	0.1480	0.1163
	(0.6173)	(0.3306)	(0.3220)	(0.2499)
ROA	0.0440	0.0494	0.0538	0.0428
	(0.0502)	(0.0566)	(0.0617)	(0.0498)
Tobin Q	−0.2657 ***	−0.2570 ***	−0.2571 ***	−0.2569 ***
	(−3.8107)	(−3.7176)	(−3.7200)	(−3.8077)
Cash	−0.0135 *	−0.0128 *	−0.0130 *	−0.0126 *
	(−1.8532)	(−1.7426)	(−1.7720)	(−1.7337)
_Cons	16.8779 ***	15.5013 ***	15.4764 ***	15.3399 ***
	(23.5245)	(21.1402)	(21.1096)	(20.4120)
Industry	Yes	Yes	Yes	Yes
*N*	1779	1774	1774	1774
Adj. *R*^2^	0.0261	0.0370	0.0366	0.0423

Note: *t* statistics in parentheses. * *p* < 0.10, ** *p* < 0.05, *** *p* < 0.01.

**Table 5 ijerph-19-11391-t005:** Regression of instrumental variables (2SLS).

	(1)	(2)
Variables	First Stage	Second Stage
	Degree	Env
Degree		1.2676 ***
		(3.1814)
Degree instrument	−0.3398 ***	
	(−5.9104)	
Dual	−3.0524 ***	3.7648 ***
	(−41.3572)	(2.7929)
Dual × Degree	0.9043 ***	−1.0757 ***
	(51.3854)	(−2.6925)
Gender	0.2053 ***	0.2743
	(3.0630)	(1.2171)
Age	−0.0064 **	0.0471 ***
	(−1.9702)	(4.7276)
Manu	−0.3734 ***	0.6349 *
	(−7.6438)	(1.6682)
SOE	0.0688 **	0.0956
	(1.9883)	(0.9607)
Dispersion	−0.0758	0.5818
	(−0.4157)	(1.2165)
Lev	0.3060 ***	2.5794 ***
	(3.6420)	(9.6272)
ROA	−0.1879	4.7992 ***
	(−0.6424)	(6.5125)
Tobin Q	−0.0405 ***	−0.3485 ***
	(−2.6325)	(−7.8654)
Cash	−0.0008	0.0214 **
	(−0.2293)	(2.2032)
_Cons	3.8906 ***	8.0616 ***
	(22.6234)	(4.5086)
Industry	Yes	Yes
*N*	2649	2649
Adj. *R*^2^	0.2061	0.0666

Note: *t* statistics in parentheses. * *p* < 0.10, ** *p* < 0.05, *** *p* < 0.01.

**Table 6 ijerph-19-11391-t006:** Shea’s partial R-squared.

Variable	Shea’s Partial R-sq.	Shea’s Adj. Partial R-sq.
Degree	0.0170	0.0076

**Table 7 ijerph-19-11391-t007:** Regression results: alternative measure of the CEO degree.

Variables	Panel A: Regression Results for the Full Sample			Panel B: Regression Results for the Heavy Pollution Companies Sample		
	Coefficient (*t*-Statistic)	Coefficient (*t*-Statistic)	Coefficient (*t*-Statistic)	Coefficient (*t*-Statistic)	Coefficient (*t*-Statistic)	Coefficient (*t*-Statistic)
	(1)	(2)	(3)	(1)	(2)	(3)
	Env	Env	Env	Env	Env	Env
Degree 2	0.5221 **	0.5313 **	0.6921 ***	0.5520 **	0.5566 **	0.8266 ***
	(2.5356)	(2.5767)	(3.0570)	(2.2506)	(2.2705)	(2.9903)
Dual		−0.1384	0.8043 *		−0.0682	1.1858 **
		(−0.9172)	(1.8240)		(−0.4052)	(2.5334)
Degree 2× Dual			−1.0704 **			−1.4285 ***
			(−2.3981)			(−2.9604)
Gender	0.0071	−0.0105	−0.0362	0.0766	0.0697	0.0265
	(0.0223)	(−0.0334)	(−0.1155)	(0.1726)	(0.1570)	(0.0605)
Age	0.0255 ***	0.0275 ***	0.0263 ***	0.0255 **	0.0264 **	0.0246 **
	(2.7387)	(2.8210)	(2.6846)	(2.3280)	(2.3255)	(2.1470)
Manu	−0.5669**	−0.5798 **	−0.4407*	0.0000	0.0000	0.0000
	(−2.1408)	(−2.1976)	(−1.6797)	(.)	(.)	(.)
SOE	−0.4793 *	−0.4873 *	−0.4232 *	−1.0216 ***	−1.0271 ***	−0.9064 ***
	(−1.8932)	(−1.9334)	(−1.7267)	(−3.4358)	(−3.4896)	(−3.2644)
Dispersion	0.6449	0.6360	0.5535	−0.9185	−0.9252	−1.0819
	(0.6170)	(0.6068)	(0.5269)	(−0.8156)	(−0.8199)	(−0.9585)
Lev	0.4395	0.4294	0.3461	0.2433	0.2387	0.0913
	(1.0494)	(1.0319)	(0.8241)	(0.5264)	(0.5195)	(0.1935)
ROA	−0.8793	−0.8734	−0.9795	0.1055	0.1115	−0.0444
	(−1.0868)	(−1.0825)	(−1.2498)	(0.1212)	(0.1281)	(−0.0526)
Tobin Q	−0.1385 **	−0.1388 **	−0.1380 **	−0.2589 ***	−0.2589 ***	−0.2583 ***
	(−2.2257)	(−2.2377)	(−2.2500)	(−3.7456)	(−3.7487)	(−3.8809)
Cash	−0.0111 *	−0.0114 *	−0.0115 *	−0.0131 *	−0.0133 *	−0.0136 *
	(−1.7222)	(−1.7614)	(−1.7805)	(−1.7561)	(−1.7866)	(−1.8316)
_Cons	15.7214 ***	15.6744 ***	15.5090 ***	16.1828 ***	16.1545 ***	16.0490 ***
	(23.3618)	(23.2979)	(23.0794)	(23.2182)	(23.1053)	(22.9156)
Industry	Yes	Yes	Yes	Yes	Yes	Yes
*N*	2660	2660	2660	1779	1779	1779
Adj. *R*^2^	0.0199	0.0201	0.0248	0.0344	0.0341	0.0441

Note: *t* statistics in parentheses. * *p* < 0.10, ** *p* < 0.05, *** *p* < 0.01.

**Table 8 ijerph-19-11391-t008:** Regression results: manu- and non-manu-industry differences.

	Manu-Industry	Non-Manu-Industry	Manu-Industry	Non-Manu-Industry
	(1)	(2)	(3)	(4)
	Env	Env	Env	Env
Degree	0.1748 **	0.3728 **		
	(2.1940)	(2.1290)		
Degree 2			0.4371 **	1.6352 **
			(2.2036)	(2.3725)
Dual	−0.0356	−0.9080 ***	−0.0429	−0.9359 ***
	(−0.2131)	(−3.2962)	(−0.2628)	(−3.4140)
Gender	0.0162	−0.3984	0.0194	−0.8299
	(0.0478)	(−1.0114)	(0.0557)	(−1.5035)
Age	0.0297 ***	0.0285	0.0272 **	0.0261
	(2.7433)	(1.3430)	(2.4519)	(1.3407)
SOE	−0.7823 ***	0.2953	−0.9265 ***	0.3419
	(−2.6336)	(1.2060)	(−3.1238)	(1.2586)
Dispersion	−0.8737	7.5246 ***	−0.9613	7.9465 ***
	(−0.8874)	(2.6804)	(−0.9906)	(2.6965)
Lev	0.4933	−0.5474	0.5229	−0.6368
	(1.1779)	(−0.4398)	(1.2788)	(−0.5049)
ROA	−0.6680	−1.5134	−0.6250	−2.1526
	(−0.9064)	(−0.7280)	(−0.8489)	(−1.1668)
Tobin Q	−0.1888 ***	0.0328	−0.1848 ***	0.0729
	(−3.0148)	(0.2624)	(−2.9612)	(0.5576)
Cash	−0.0100	−0.0125	−0.0107	−0.0152
	(−1.3860)	(−0.9185)	(−1.4725)	(−1.1574)
_Cons	15.1550 ***	14.0921 ***	15.5672 ***	14.3659 ***
	(21.7813)	(8.8707)	(23.9604)	(11.2621)
Industry	Yes	Yes	Yes	Yes
*N*	2048	601	2057	603
Adj. *R*^2^	0.0266	0.0389	0.0279	0.0556

Note: *t* statistics in parentheses. ** *p* < 0.05, *** *p* < 0.01.

**Table 9 ijerph-19-11391-t009:** Regression results: high-market competition and low-market competition.

Variables	Panel A: Full Sample		Panel B: Low-Market Competition Sample	Panel C: High-Market Competition Sample
			High-Monopoly/Low-Market Competition	Low-Monopoly/High-Market Competition
			Lerner Index > Mean of Lerner Index	Lerner Index < Mean of Lerner Index
	(1)	(2)	(3)	(4)
	Env	Env	Env	Env
Degree		0.1972 ***	0.2898 ***	0.2082 **
		(2.6533)	(2.9434)	(2.1636)
Lerner Index	2.6145 **	2.5836 **		
	(2.1215)	(2.1381)		
Dual	−0.0885	−0.1194	0.0703	−0.1551
	(−0.6193)	(−0.7995)	(0.2596)	(−0.7995)
Gender	−0.0012	−0.0231	−0.0034	0.0076
	(−0.0041)	(−0.0773)	(−0.0068)	(0.0219)
Age	0.0214 **	0.0275 ***	0.0131	0.0279 **
	(2.2424)	(2.8472)	(0.8193)	(2.2573)
Manu	−0.5722 **	−1.6354 ***	−1.4294 ***	1.3753 ***
	(−2.2279)	(−6.4648)	(−2.9797)	(5.3568)
SOE	−0.4091 *	−0.3462	−0.2203	−0.4662 *
	(−1.6599)	(−1.4336)	(−0.4982)	(−1.6660)
Dispersion	0.5352	0.6691	−0.6362	0.2842
	(0.5257)	(0.6307)	(−0.3084)	(0.2018)
Lev	0.5255	0.4615	0.4226	−0.1386
	(1.2511)	(1.0997)	(0.4756)	(−0.3002)
ROA	−1.2300	−1.2376	−3.1913 *	−0.4400
	(−1.5409)	(−1.5204)	(−1.7838)	(−0.5101)
Tobin Q	−0.1337 **	−0.1346 **	−0.1069	−0.1246 *
	(−2.1900)	(−2.1833)	(−0.9120)	(−1.7676)
Cash	−0.0099	−0.0094	−0.0050	−0.0114
_Cons	16.0384 ***	15.9186 ***	16.4501 ***	13.8386 ***
	(23.7181)	(22.6100)	(15.3043)	(16.8643)
Industry	Yes	Yes	Yes	Yes
*N*	2660	2649	909	1740
Adj. *R*^2^	0.0161	0.0222	0.0237	0.0108

Note: *t* statistics in parentheses. * *p* < 0.10, ** *p* < 0.05, *** *p* < 0.01.

## Data Availability

In this study, we used data from the China Stock Market & Accounting Research (CSMAR) Database. For data availability, please contact the authors.

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
