# Peer review of "Is Education Beneficial to Environmentally Friendly Behaviors? Evidence from CEOs"

_ijerph, 2022, doi:10.3390/ijerph191811391_

Round 1
Reviewer 1 Report
Both the topic and the methods are interesting and meaningful since CEO is a group of important actors in environmental protection. The Language needs to be refined to avoid repetition and looks more concise.
1. Strength:
1) The academic meaning of this manuscript is to highlight the importance of the CEOs education level on environmental protection investment, which might help the government or enterprises to choose better CEO, especially for the non-manufacture industry.
2) It is innovative since it is really challenging to collect all the available data on this research and establish the modelling.
2. Weakness:
1) In the last paragraph of P3, They view CRS activities should be They view CSR activities.
2) In the last paragraph of P5, CAHS TO PROFIT should be CASH TO Profit.
3) In the last paragraphs of P16, It is better for the author to be more logic and streamline.
4) In the last paragraph of P16, On the other hand seems redundant.
Author Response
Thank you very much for your valuable comments on our paper, which will play a critical role in improving the quality of the paper. We have carefully answered the questions according to the comments and carefully revised the article. The article's revisions are highlighted using the “Track Changes” function. Thanks again for your help and guidance with this article.
Comment 1: In the last paragraph of P3, They view CRS activities should be They view CSR activities.
Response1: The CRS activities have been modified to CSR activities in lines 169-171 in the last paragraph of Page 4.
Comment2: In the last paragraph of P5, CAHS TO PROFIT should be CASH TO Profit.
Response 2: The “CAHS TO PROFIT” has been modified to “CASH TO Profit” in line 296 in the last paragraph of Page 6.
Comment 3: In the last paragraphs of P16, It is better for the author to be more logic and streamline.
Response 3: We have reorganized the last paragraphs of Page 18 in lines 218-225.
Comment 4: In the last paragraph of P16, On the other hand seems redundant.
Response 4: “On the other hand” in line 223 of the last paragraph of Page 18 is deleted.
Reviewer 2 Report
The manuscript presents an interesting study to investigate the relationship between CEO education and corporate environmental protection investment, based on a panel sample of Chinese listed firms from 2010 to 2019. The authors discuss how CEO duality, manufacturing, and monopoly influence the relationship and partly reveal its economic mechanism. This study provides useful information on corporate environmental activities from the perspective of CEO education. The manuscript fits the scope of the International Journal of Environmental Research and Public Health and is worth publishing. I have some minor comments below for the authors to address before its acceptance.
(1) Clearly describe the innovations of this study in the Introduction section.
(2) There is too little relevant literature (about environmental protection investment and CEO education, etc) in the literature review, which needs to be more detailed.
(3) Explain what "CEO duality" is, when it first appears (line 79).
(4) Give the full name of 2SLS (line 87) when it first appears, and use abbreviation in the rest of this article (Section 4.4).
(5) Add the full name of CSR and CRS (lines 137-144).
(6) "Table 1" (line 264) and "Table 8" (line 132) are wrongly numbered.
(7) Write concisely and avoid redundant information. For example, "The moderating effect model based on CEO duality shows that CEO duality can be used as..." could be refined as "CEO duality can be used as...".
Author Response
Thank you very much for your valuable comments on our paper, which will play a critical role in improving the quality of the paper. We have carefully answered the questions according to the comments and carefully revised the article. The article's revisions are highlighted using the “Track Changes” function. Thanks again for your help and guidance with this article.
Comment 1: Clearly describe the innovations of this study in the Introduction section.
Response 1: We have described the innovations of this paper in the introduction and it is described in lines 113-130 of Page 3.
Comment 2: There is too little relevant literature (about environmental protection investment and CEO education, etc) in the literature review, which needs to be more detailed.
Response 2: We have detailed the relevant literature, and it is shown in lines 152-162 of Page 4 and lines 206-221 of Page 5.
Comment 3: Explain what "CEO duality" is, when it first appears (line 79).
Response 3: “CEO duality” has been explained as the CEO also holds the position of board chairman when it first appears in line 80 of Page 2.
Comment 4: Give the full name of 2SLS (line 87) when it first appears, and use abbreviation in the rest of this article (Section 4.4).
Response 4: The full name of two-stage least squares is given in line 90 of Page 2. The abbreviation is applied in section 4.4, and it is shown in line 60 in Page 12.
Comment 5: Add the full name of CSR and CRS (lines 137-144).
Response 5: The full name of CSR is shown as corporate social responsibility in line 164 and the CRS is changed to CSR in lines 169-171 of Page 4.
Comment 6: "Table 1" (line 264) and "Table 8" (line 132) are wrongly numbered.
Response 6: “Table 1” has been modified to “Table 2” in line 317 of Page 7 and “Table 8” has been modified to “Table 9” in line 133 of Page 15.
Comment 7: Write concisely and avoid redundant information. For example, "The moderating effect model based on CEO duality shows that CEO duality can be used as..." could be refined as "CEO duality can be used as...".
Response 7: “The moderating effect model based on CEO duality shows that CEO duality can be used as a moderating variable between CEO education and firms' environmental protection investment behaviors” has been changed to “CEO duality can be used as a moderating variable between CEO education and firms' environmental protection investment behaviors” in lines 165-166 of Page 17.